# Disconnecting from Difficult Emotions in Times of Crisis: The Role of Self-Compassion and Experiential Avoidance in the Link Between Perceived COVID-19 Threat and Adjustment Disorder Severity

**DOI:** 10.3390/healthcare13080934

**Published:** 2025-04-18

**Authors:** Paweł Holas, Aleksandra Juszczyk, Jan Wardęszkiewicz, Joseph Ciarrochi, Steven C. Hayes

**Affiliations:** 1Faculty of Psychology, University of Warsaw, 00-183 Warsaw, Poland; a.juszczyk@uw.edu.pl (A.J.); jan.wardeszkiewicz@psych.uw.edu.pl (J.W.); 2Institute for Positive Psychology and Education, Australian Catholic University, Brisbane 4014, Australia; ciarrochij@gmail.com; 3Department of Psychology, University of Nevada, Reno, NV 89154, USA; hayes@unr.edu

**Keywords:** adjustment disorder, experiential avoidance, self-compassion, perceived threat of COVID-19, depression, anxiety

## Abstract

**Objectives:** The COVID-19 pandemic has significantly impacted mental health worldwide. This study investigated the relationship between perceived COVID-19 threat and adjustment disorder (AjD) severity, examining self-compassion (SC) and experiential avoidance (EA) as potential moderators. Additionally, cluster analysis—a statistical method for grouping individuals based on similar psychological characteristics—was employed to identify distinct profiles of SC and EA and their associations with AjD, depression, and anxiety symptoms. **Methods:** A sample of 308 participants meeting AjD criteria completed measures assessing AjD severity (ADNM-20), depression (PHQ-9), anxiety (GAD-7), SC, EA, and perceived threat of COVID-19. Moderation analyses were performed using the PROCESS macro. Cluster analysis identified profiles based on SC and EA scores, with clusters compared on AjD, PHQ, and GAD symptom severity. **Results:** SC and EA moderated the relationship between perceived COVID-19 threat and AjD severity. Interestingly, individuals with high EA and low SC exhibited no significant association between perceived threat and AjD symptoms. Cluster analysis revealed four distinct profiles: (1) high SC and low EA, (2) average SC and EA, (3) low SC and low EA, and (4) low SC and high EA. Participants in the high SC/low EA cluster reported significantly lower levels of AjD, depression, and anxiety symptoms compared to those in the low SC/high EA cluster, who exhibited the highest symptom severity across all measures. **Conclusions:** Our findings suggest that individuals who relied on experiential avoidance and lacked self-compassion experienced less emotional distress related to pandemic-related worries, potentially shielding themselves from acute AjD symptoms. However, this strategy was associated with greater emotional distress, as those with high AE and SC exhibited more symptoms of AjD, depression, and anxiety. In contrast, individuals with low AE and high SC demonstrated significantly better psychological well-being.

## 1. Introduction

Adjustment disorder (AjD), characterized by a maladaptive response to a significant life stressor occurring within one month of the event, typically includes a preoccupation with the stressor and difficulties in adapting [1]. It is one of the most common mental health diagnoses in clinical practice [2], yet remains relatively understudied [3]. Despite this, there has been a growing interest in AjD, especially following the release of clear diagnostic criteria for the disorder in the 11th edition of the International Classification of Diseases (ICD-11) [4]. Life stressors that trigger AjD often involve non-traumatic but significant events, such as illness, disability, or socioeconomic difficulties [5]. The COVID-19 pandemic, which began in March 2020, became a major stressor with profound mental health consequences, including a rise in the prevalence of AjD [6,7]. Factors such as self-isolation, quarantine, job loss, and perceived risk of contracting COVID-19 were identified as key risk factors for developing AjD [8]. Indeed, international health guidelines recognized quarantine as a potentially distressing experience with psychological consequences, particularly if prolonged [9].

Given the pandemic’s global and prolonged nature, it represented a unique type of stressor—ubiquitous, ongoing, and often ambiguous—which aligns closely with the kinds of stressors that trigger AjD. While a lot of research has focused on PTSD, anxiety, and depression during the pandemic, adjustment disorder offers a more context-specific framework for understanding short-to-mid-term psychological maladaptation in response to such diffuse life changes. The shifting threat landscape, uncertain health and economic conditions, and widespread disruption of daily life created conditions that mirror the diagnostic profile of AjD, namely a preoccupation with the stressor and significant difficulty adjusting to it. At the same time, certain behavioral and cognitive strategies, such as adaptive social comparisons or online engagement, may have helped buffer psychological well-being during lockdowns [10]. Therefore, AjD may offer a uniquely relevant lens for understanding mental health vulnerability during the COVID-19 crisis.

A central feature of the pandemic was the heightened perception of threat to personal and collective safety [11], with studies indicating that a high perceived threat level was strongly correlated with poorer mental health outcomes [12]. Specifically, perceptions of threat have been linked to higher levels of anxiety and depression [13,14]. Given the detrimental impact of perceived threat on stress response and coping mechanisms [15], it is unsurprising that the perceived threat of the COVID-19 pandemic has emerged as a significant predictor of AjD. For instance, a Polish study during the pandemic’s early phase found that 75% of participants considered COVID-19 a highly stressful event, which was the strongest predictor of AjD. Additionally, 49% reported increased AjD symptoms, with higher prevalence among females and those without full-time employment [16]. The European Society for Traumatic Stress Studies (ESTSS) conducted the ADJUST Study, involving 15,563 adults across eleven European countries, which found a prevalence of self-reported probable adjustment disorder (AjD) of 18.2% [17]. Pandemic-related stressors associated with higher levels of AjD symptoms included, among others, fear of infection. These findings underscore the important role of perceived threat in the development of AjD during the COVID-19 pandemic. However, only a small amount of research has explored the moderators of the relationship between perceived threat and AjD symptoms.

Therefore, in the current study we wanted to evaluate whether self-compassion (SC) and experiential avoidance (EA) act as moderators of this relationship. Both SC and EA have been shown to play crucial roles in emotional regulation, and in this study we explored whether they serve as moderators—variables that influence the strength or direction of the relationship—between perceived threat of COVID-19 (hereafter referred to as “perceived threat”) and AjD symptoms. Both high SC and low EA (or, conversely, high experiential acceptance) were found to be protective factors against mental health difficulties [18].

Self-compassion involves treating oneself with kindness during times of suffering, recognizing one’s shared humanity, and maintaining a balanced, non-judgmental awareness of difficult emotions [19]. Research has consistently demonstrated that SC is associated with improved resilience and well-being [20], acting as a buffer in high-stress situations [21]. Self-compassion has been shown to moderate the effects of stress on mental health outcomes [22]. More recently, a study conducted during the pandemic found that self-compassion helped to mitigate the negative psychological impact of COVID-19 threat [23]. This suggests that SC could reduce the emotional toll of perceived health threats and consequently decrease the severity of AjD. Intriguingly, not all previous research showed a mitigating role of self-compassion to stress, as other studies have found different moderation patterns [24,25]. In this respect, Dev et al. [24] showed the opposite effects, with the association between stress and burnout being stronger, not lesser, in nurses with greater self-compassion. It opens the possibility that the role of self-compassion can be more nuanced, depending on the context and study population.

Experiential avoidance (EA) refers to the tendency to avoid or suppress distressing thoughts, emotions, or memories, even when such avoidance leads to negative long-term outcomes [26]. EA is closely tied to psychological inflexibility and has been associated with poorer outcomes following stress exposure [27]. High EA is linked to a variety of mental health conditions, including depression and anxiety, e.g., ref. [28], and acts as a mediator between stress exposure and negative mental health consequences [29]. In the context of the pandemic, individuals with high EA may struggle more to adapt to stressors and are more likely to develop mental health issues, including AjD.

Although SC and EA are negatively correlated [30], both are important constructs to study in parallel because they represent distinct-but-complementary aspects of emotional regulation. Self-compassion involves actively embracing distress, whereas experiential avoidance involves attempting to escape or suppress it. Understanding how these two constructs interact could provide valuable insights into how individuals cope with stressful life events such as the pandemic. Moreover, previous work has highlighted that combining these two constructs can help in understanding the full spectrum of emotional responses to stress [31]. Given that these processes may moderate the relationship between the COVID-19 threat and AjD, this study aims to explore their potential interactive effects.

In the present study, we measure not only adjustment disorder (AjD) but also the intensity of depression and anxiety, as these conditions are often comorbid with AjD and share common underlying mechanisms, such as maladaptive emotional regulation [3]. Depression and anxiety are frequently observed alongside AjD [3], and by measuring them together we aim to provide a broader understanding of how perceived threat and emotional regulation interact to affect mental health. Given the high prevalence of these conditions during the COVID-19 pandemic [13,14], incorporating them into this study allows for a more nuanced understanding of how self-compassion (SC) and experiential avoidance (EA) are related not only to adjustment symptoms but also to broader psychological distress. Both self-compassion and experiential avoidance can function as moderators of the relationship between perceived threat and AjD symptoms. Additionally, their configuration could be linked to varying levels of psychopathological symptoms, suggesting that different combinations of SC and EA may influence the severity of depression, anxiety, and AjD symptoms.

The current study evaluates the associations between self-compassion, experiential avoidance, perceived threat, and psychopathological symptoms in a sample of individuals with AjD. Specifically, we test whether high levels of SC and low levels of EA attenuate the effect of perceived COVID-19 threat on AjD symptom intensity. Furthermore, we explore whether the intensity of depression and anxiety symptoms differs across clusters of individuals with varying levels of EA and SC. Based on prior work, we hypothesize that high SC and low EA (experiential acceptance) will weaken the association between perceived threat and AjD symptom severity. Additionally, we hypothesize that individuals with AjD characterized by a high level of EA and low level of SC will exhibit significantly higher symptoms of AjD, depression, and anxiety compared to those with the opposite profile, i.e., low EA (experiential acceptance) and high SC.

## 2. Materials and Methods

### 2.1. Recruitment

This study was conducted in June 2020 and was part of a larger project aimed at the evaluation of the effectiveness of online mindfulness therapeutic intervention for people experiencing adjustment disorder due to the COVID-19 pandemic. Participants were recruited via the Internet. Advertisements were posted on Facebook’s psychological support groups, psychological fan pages promoting well-being, students’ groups, and Instagram’s lifestyle, and the invitations to this study were sent in some university newsletters.

On the dedicated platform www.covid.stress-less.pl (accessed on 8 June 2020), as well as in the advertisements, it was explained that this study is designed for people experiencing emotional difficulties related to the COVID-19 pandemic and its consequences and that registering for this study did not guarantee participation in the intervention since an individual may not fulfill the inclusion criteria. Informed consent was mandatory for participation in this study.

### 2.2. Participants

The webpage initially registered 790 people, of which 564 filled in all the obligatory screening questionnaires. Participants were selected if they met the inclusion criteria: having a diagnosis of AjD (higher than the cut-off score (47.5) in ADNM-20 [30]) and meeting the criteria of emotional disorder (a cut-off score of ≥8 for both scales (anxiety and depression)) on HADS [31]. These criteria led to a final sample of 308 participants. Characteristics of the sample are shown in Table 1.

### 2.3. Measures

Background information of participants: Participants answered questions about their birth year, sex, education level, their professional and financial situation, marital status, the current level of their socioeconomic status related to the COVID-19 pandemic, and any indication of the use of therapeutic/pharmacological methods in the past or present.

Adjustment disorder (Adjustment Disorder New Module-20, ADNM-20 [32]): The ADNM-20 questionnaire is used to assess adjustment disorder. It consists of two parts: a stressor list (which includes a range of acute and chronic life events from the past two years) and an item list (which evaluates the symptoms in response to the most distressing events). Participants responded on a Likert-type scale about how often they have experienced adjustment disorder symptoms in the past two weeks (from 0—never to 3—often). The ADNM-20 questionnaire consists of six subscales (preoccupation, failure to adapt, avoidance, depressive mood, anxiety, and impulse disturbance), with preoccupation and failure to adapt as core symptoms, and avoidance, depressive mood, anxiety, and impulse disturbance as accessory symptoms of adjustment disorder diagnosis.

The internal consistency of this questionnaire is high (Cronbach’s α = 0.94), as well as the core symptoms summed in one scale (Cronbach’s α = 0.90) and separately (Cronbach’s α = 0.88 for preoccupation and Cronbach’s α = 0.80 for failure to adapt). The subscale for accessory symptoms also showed a high internal consistency (Cronbach’s α = 0.89). Test–retest reliability was not reported in this study.

Depression (Patient Health Questionnaire-9; PHQ-9 [33]; Polish adaptation [34]): PHQ-9 is used to assess the level of depressive symptoms. It consists of nine statements derived from the DSM-IV criteria of depressive disorder and an additional statement regarding the severity of existing symptoms in daily life. If participants indicated any of the symptoms, they had to answer this question on a scale from 1 (definitely not related) to 5 (definitely related). The PHQ-9 in the current study showed a good internal consistency (Cronbach’s α = 0.83). Test–retest reliability has been established in prior research but was not assessed in the present study.

Anxiety (Generalized Anxiety Disorder Scale-7, GAD-7 [35]): GAD-7 is a measure used to assess the level of anxiety symptoms in generalized anxiety disorder (GAD) defined by the Diagnostic and Statistical Manual of Mental Disorders, Fourth Edition (DSM-IV). It is a scale that contains seven items that relate to characteristics of GAD (feeling anxious, worrying too much, having difficulties relaxing, etc.). The psychometric properties of this questionnaire are strong. The GAD-7 demonstrated excellent psychometric properties. Internal consistency was high (Cronbach’s α = 0.92), and test–retest reliability, assessed via intraclass correlation, was also strong (ICC = 0.83).

Anxiety and depression (Hospital Anxiety and Depression Scale; HADS [36]; Polish adaptation [37]): HADS is a self-report measure used to assess depressive and anxiety symptoms. The questionnaire is built up of 14 items: 7 are related to anxiety, e.g., “I feel tense or wound up”, and 7 are related to depression, e.g., “I look forward with enjoyment to things”. Factor analyses of the two subscales show a two-factor solution in good correspondence with the HADS subscales for Anxiety (HADS-A) and Depression (HADS-D), respectively. Cronbach’s alpha for HADS-A varies from 0.68 to 0.93 (mean 0.83) and for HADS-D from 0.67 to 0.90 (mean 0.82).

Self-compassion (Self-Compassion Scale Short Form, SCS-SF [38]; Polish adaptation [39]): Self-Compassion Scale consists of 26 items, which are grouped into six subscales: self-kindness, self-judgment, common humanity, isolation, mindfulness, and over-identification. The internal consistency of the questionnaire is high (Cronbach α = 0.92.), and test–retest reliability was acceptable (r = 0.93 for the overall score).

Perceived Health and Life Risk of COVID-19 scale (PHLRC, 11): The scale consists of six questions assessing the subjective risk of COVID-19 infection, serious adverse health effects and complications due to a coronavirus infection, and a threat to life as a result of an infection. Each of these areas was assessed using two items: one relating to oneself and a second one relating to loved ones. The six items were rated on a five-point scale from 1—very low to 5—very high. Cronbach’s alpha was α = 0.92. Test–retest reliability was not reported.

Experiential avoidance (Acceptance and Action Questionnaire, AAQ-II [40]): AAQ-II is the most widely used measure of psychological inflexibility and experiential avoidance. Lower scores on AAQ-II are also indicators of psychological flexibility. AAQ-II consists of seven items (e.g., “I am afraid of my feelings”, “I worry about not being able to control my worries and feelings”) rated from 0 (never true) to 7 (always true) on an eight-point Likert scale. Developed originally as a measure of psychological inflexibility, its items tend to emphasize experiential avoidance and some have suggested that it be interpreted that way [41]. In the current study, Cronbach’s alpha was α = 0.90. Test–retest reliability has been established in prior studies but was not directly measured in this sample.

### 2.4. Statistical Analyses

We began by calculating descriptive statistics and Pearson correlation coefficients among the study variables. Moderation analyses were then conducted to examine whether experiential avoidance (EA) and self-compassion (SC) influenced the strength of the relationship between perceived COVID-19 threat (PHLRC) and adjustment disorder severity (ADNM-20). These analyses were carried out using Hayes’ PROCESS macro for SPSS (version 3.5.3), employing Model 1 to test individual moderation effects and Model 3 to test for potential interaction between moderators [42]. All predictors were mean-centered prior to analysis to facilitate interpretation. Statistical significance of moderation was assessed through 95% confidence intervals (CIs) derived via bootstrapping (5000 samples). The Johnson–Neyman technique was applied to identify regions of significance for the interaction effects. The sample in the current study was equal to 308 participants. Assuming the statistical power to be 0.8 and planning to perform moderation analysis with one moderator and one explaining variable in a single statistical model, one needs to encounter the interaction effect equal to at least 0.03 in terms of Cohen’s f2 effect size measure to detect it as statistically significant. According to Cohen [43] (1988), an effect size of f2 = 0.02 is to be considered small. Therefore, the sample in the present study is not sensitive enough to detect it. The moderation effect needs to be a bit stronger. As a result of the limited statistical power, we formulated hypotheses of the attenuation effect instead of the buffering effect. In Model 3, we evaluated the three-way interaction between PHLRC, SC, and EA to determine whether the moderating effects of SC and EA were conditional on one another. The Johnson–Neyman procedure was applied to identify the range of moderator values where the conditional effect of the predictor (PHLRC) on the outcome (ADNM-20) was statistically significant. Results were visualized with interaction plots and interpreted based on standardized coefficients. To examine heterogeneity in psychological profiles, a k-means cluster analysis was conducted using standardized scores of AAQ-II (EA) and SCS-SF (SC) as input variables. The analysis was performed using IBM SPSS Statistics 28.0. A four-cluster solution was chosen based on theoretical assumptions and supported by visual inspection of the within-cluster sum of squares. The goal was to classify participants into distinct psychological profiles representing combinations of high/low SC and EA. The final cluster solution comprised the following: 1. high SC/low EA; 2. average SC/average EA; 3. low SC/low EA; and 4. low SC/high EA. A four-cluster solution led to the extraction of four groups that were balanced in terms of the number of participants. Following cluster extraction, a one-way ANOVA was conducted to compare levels of AjD, depression (PHQ-9), and anxiety (GAD-7) symptoms across the four groups. Post hoc comparisons were conducted using the Games–Howell test to account for heterogeneity of variance. Effect sizes (η^2^) were computed for each outcome to assess the magnitude of between-cluster differences.

## 3. Results

### 3.1. Descriptive Statistics

Descriptive statistics and Pearson correlation coefficients between analyzed variables can be found in Table 2.

The ADNM-20 showed positive correlations with the symptom intensity scales (i.e., PHQ-9, GAD-7, HADS) and with the AAQ-II, but a negative correlation with SCS-SF. The symptom intensity scales were positively correlated with each other. Additionally, the AAQ-II correlated positively with all of the symptom intensity scales. SCS-SF correlated negatively with all symptom intensity scales and AAQ-II. A positive correlation was observed between perceived health risk (PHLRC) and both depressive and anxiety symptoms, suggesting that individuals perceiving greater threat also reported more intense psychological distress. No statistically significant correlation was found between ADNM-20 and PHLRC; however, we hypothesized a moderating effect between these variables, so a lack of correlation in the total sample was expected.

Regarding sociodemographic variables, we refrained from analyzing gender differences due to the overrepresentation of female participants. No significant correlation was found between age and the intensity of ADNM symptoms, r(306) = 0.02, *p* > 0.05. Similarly, participants with higher education did not differ significantly in ADNM symptom severity compared to those without higher education, *t*(306) = 0.98, *p* > 0.05.

### 3.2. AAQ and Self-Compassion as Moderators of the Relationship Between PHRLC and ANDM

The results of the moderation analysis are shown in Table 3.

AAQ and self-compassion were analyzed as moderators of the relationship between PHLRC and ADNM-20. Consistent with the correlation analysis, no main effect of AAQ on PHLRC was found, although self-compassion positively correlated with PHLRC. Importantly, both interaction effects were statistically significant, indicating moderation effects. The three-way interaction was not statistically significant, B = [−0.09; 0.07], *p* > 0.05. To interpret these interactions further, a Johnson–Neyman procedure was applied, and the results are presented in Table 4.

The findings revealed that a positive relationship between PHLRC and ADNM was statistically significant when AAQ levels were low (below the mean) and SCS levels were high (above the mean), as depicted in Figure 1 and Figure 2.

### 3.3. Profiles of AAQ and Self-Compassion and Their Relationships with Psychological Symptom Intensity

Next, we examined the relationship between AAQ and self-compassion profiles and the intensity of ADNM, PHQ, and GAD symptoms. To identify these profiles, participants’ scores on AAQ and SCS-SF were analyzed using k-means cluster analysis. Four clusters were extracted, as described in the rationale. A one-way ANOVA showed significant differences between the clusters in terms of both AAQ scores, F(3, 304) = 313.53, *p* < 0.001, and SCS-SF scores, F(3, 304) = 264.54, *p* < 0.001. The first cluster (n = 40) was characterized by a low level of AAQ and a high level of SCS-SF (see Figure 3).

The second cluster (n = 93) was characterized by average levels of both AAQ and SCS-SF. The third cluster (n = 77) was characterized by lower levels of both AAQ and SCS-SF. The fourth cluster (n = 98) was characterized by a high level of AAQ and a low level of SCS-SF. Clusters 1, 2, and 4 were consistent with the negative correlation between AAQ and SCS-SF (see Table 2), but cluster 3 deviated from this pattern, reflecting the average strength of this correlation. We then compared the extracted clusters in terms of PHQ, GAD, and ADNM symptom intensity. The results are presented in Table 5, supplemented by Figure 4 for clarity.

Statistically significant differences were found between the clusters in terms of AjD (ADNM-20), GAD (GAD-7), and depressive (PHQ-9) symptom intensity. According to the values of the Games–Howell post hoc test, ADNM intensity in cluster 4 was significantly higher than in cluster 1, *p* < 0.001, and 3, *p* < 0.001. Also, AjD intensity was significantly higher in cluster 2 than in cluster 1, *p* < 0.01. The intensity of GAD symptoms in cluster 4 was significantly higher than in clusters 1, *p* < 0.001, 2, *p* < 0.001, and 3, *p* < 0.001. The intensity of depressive symptoms in cluster 4 was also significantly higher than in cluster 1, *p* < 0.001, 2, *p* < 0.001, and 3, *p* < 0.001.

## 4. Discussion

Our aim in the current study was to investigate the associations between self-compassion, experiential avoidance, perceived threat, and psychopathological symptoms in a sample of individuals with AjD. To begin with, our results revealed a statistically significant positive association between perceived threat and depression and anxiety symptoms. This finding is in line with previous studies conducted during the COVID-19 pandemic that found a link between increased levels of the pandemic threat and psychopathology, including anxiety, depression, and negative affect, e.g., refs. [12,13]. Broadly speaking, it is also congruent with the general view that it is not an objective threat, but the subjective perception of a threat, that is related to mental well-being [14]. No statistically significant correlation was found between perceived COVID-19 threat and AjD severity. Nonetheless, this is consistent with our hypothesis that the relationship between these variables would be moderated, meaning that the lack of a direct correlation in the total sample was expected. Previous research has indicated that the female gender and a younger age are associated with higher levels of AjD symptoms [4,16,17]. Although the overrepresentation of female participants in our sample limited the feasibility of meaningful gender comparisons, we conducted an analysis to examine the relationship between age and AjD severity. However, no significant association was observed between these variables.

Turning to the moderation effects, our analysis revealed unexpected results. We hypothesized that those with a high level of SC and a low level of EA (higher acceptance) would exhibit an attenuating effect reflected in reducing or turning off the significance of this relationship. While SC and AE moderated the relationship between perceived threat and AjD symptoms, it did so contrary to our expectations. Rather than weakening the association between perceived threat and AjD symptoms, this association was stronger. Although this result is less consistent with previous evidence demonstrating the potency of self-compassion and experiential acceptance as mitigators of negative outcomes [22], it aligns with some previous findings. Dev et al. [24] demonstrated that the association between stress and burnout was stronger in those with greater self-compassion. Similarly, Kyeong [25] reported that self-compassion moderated the relationship between academic burnout and psychological well-being, but the relationship was stronger at higher levels of SC.

One possible explanation for this outcome is experiential avoidance theory [26]. This theory states that in the face of danger experiential avoidance can temporarily reduce the level of experienced distress even if its longer-term impact might be negative. This hypothesis has been confirmed in some previous studies. For example, it was found that among people characterized by high anxiety sensitivity and a high level of experiential avoidance, after experiencing the emotional evocative task the impact of this experience on negative affect is much lower than among people who have the willingness to experience distress [44]. Importantly, the moderating effect tells us nothing about the actual level of the dependent measure—in our case, AjD symptoms—and it is worth noting that participants in the present study were selected based on meeting diagnostic criteria for AjD (ADNM-20) and elevated emotional distress (anxiety and depression) on HADS. Therefore, it is possible that these individuals tended to disconnect from their emotional experiences and were harsh, rather than compassionate, toward their difficulties to prevent emotional distress triggered by worries about their own and their loved ones’ health and safety. As a result, EA and a lack of SC may have prevented emotional elaboration of pandemic-related worries, thereby helping participants to reduce the adverse emotional impact, at least temporarily. In the short term, this strategy may have “protected” them from heightened AjD symptoms during the pandemic.

A crucial question, however, is whether this protective function led to improved psychological functioning. In other words, did individuals with high EA and low SC exhibit reduced levels of depression, anxiety, and AjD symptoms compared to those with opposite characteristics (low EA and high SC)? To address this, we conducted a cluster analysis to identify participant profiles based on SCS-SF and AAQ scores and compared these profiles across symptom severity. The results were clear. Specifically, individuals with high psychological flexibility and high SC exhibited significantly lower levels of AjD, depression, and anxiety symptoms compared with those with high EA and low SC. This suggests that although avoidance and self-criticism might initially dampen emotional reactions, over time they are associated with increased psychological distress. Non-human animals are motivated to avoid negative affect by avoiding situations that produce it; humans also tend to avoid aversive private experiences [45], but since cognitive access to distressing situations is ever present, this strategy has widespread and pathological consequences over time.

Generally speaking, our findings indicate that individuals may use experiential avoidance and harsh self-judgment in an attempt to minimize distress. However, these strategies may eventually lead to worse outcomes. This interpretation is supported by previous experimental studies that found similar roles for SC and EA in the development of psychopathology [31]. Furthermore, our findings are in line with clinical observations suggesting that individuals with poor emotional insight and low self-reflection often exhibit high symptom levels and poor awareness of emotional causes [46]. The current results initially seemed to contradict empirical research indicating that psychological flexibility and self-compassion are protective factors for mental health (e.g., ref. [47]) but more detailed data analysis showed results in line with these assumptions. It is worth noting that Matos et al. [45] found that self-compassion had a moderating effect on the relationship between COVID-19 threat and emotional disorder symptoms, but there are methodological differences between that study and the present one. We invited people who experienced AjD and emotional distress symptoms due to the COVID-19 pandemic, whereas Matos at al. recruited individuals from the general population. In addition, both studies differed in terms of measures used, limiting direct comparison.

In sum, our findings support the third-wave cognitive behavioral therapy perspective, which emphasizes open awareness, acceptance, and self-kindness as mechanisms for fostering psychological resilience [48]. Consistent with this view, studies conducted during the COVID-19 pandemic have shown that mindfulness- and acceptance-based interventions can reduce the negative impact of pandemic-related stress [49,50].

### 4.1. Limitations

Several limitations need to be taken into consideration when interpreting the present findings. First of all, a recruitment announcement for this study was targeted at people seeking help for emotional distress related to the COVID-19 pandemic. Our results, therefore, may reflect the characteristics of people willing to seek help, but not the general population. In addition, the sample was characterized by a gender imbalance, with a majority of participants identifying as women. This uneven distribution may limit the generalizability of the findings, particularly in terms of their applicability to male or gender-diverse populations. Future studies should aim to recruit more gender-balanced samples or focus specifically on one gender to explore potential differences in psychological responses. Also, that the sample size was not large enough to detect a moderation effect of a small size is a limitation. It also made the precise estimation of the moderation effect size difficult. A larger sample size would allow for analyzing both moderators in a single multivariate statistical model. However, typical sample sizes in research projects evaluating adjustment disorder rarely exceed 300 people. In future research with a larger sample of participants, it would be worth running one model with two moderators in order to compare the strength of the moderation effects, and in addition to assess the possibility that one is the moderator of the moderation of the other. Secondly, our study had a cross-sectional design. Therefore, we do not know how the investigated constructs changed over time, nor what the temporal and causal relations between variables may be. Future studies need to conduct longitudinal and experimental designs that would allow for the investigation of causality effects. Finally, the investigated variables were measured with self-reported scales that are subjective and vulnerable to biases such as social desirability or the need for social enhancement [51].

### 4.2. Implication of This Research for Theory, Research, and Practice

This study offers several key implications for psychological theory, future research, and clinical practice. Theoretical Implications: Our findings support theoretical models within third-wave cognitive behavioral therapies that emphasize the importance of mindfulness, experiential acceptance, and self-compassion in emotional regulation. The results reinforce the theoretical relevance of combining emotional avoidance and compassionate self-awareness frameworks in understanding psychopathological outcomes during prolonged stress.

Research Implications: This study highlights the need for further longitudinal and experimental research to examine how these constructs interact over time, especially during ongoing global crises. Future studies should explore the temporal dynamics of SC and EA and their potential causal relationships with mental health outcomes. Additionally, the use of real-time assessment methods such as ecological momentary assessment could provide a more nuanced understanding of how these processes unfold in daily life. Expanding samples to include diverse populations and gender-balanced groups will enhance the generalizability of findings. 

Practical Implications: Clinically, these findings underscore the importance of interventions aimed at enhancing self-compassion and reducing experiential avoidance in individuals experiencing adjustment difficulties. Therapeutic approaches such as Mindfulness-Based Cognitive Therapy (MBCT) and Acceptance and Commitment Therapy (ACT) that foster psychological flexibility and compassionate self-relating may be particularly beneficial in mitigating the emotional impact of perceived threats. Promoting emotional openness and self-kindness may reduce vulnerability to AjD, especially in times of collective crisis such as a pandemic.

## 5. Conclusions

Our research provides some new knowledge about adjustment disorders and emotional distress related to the COVID-19 pandemic, as well as the mechanisms involved in their maintenance. The current results suggest that although cutting oneself off from difficult emotions in the face of danger may appear to be self-soothing, it is related to the increased likelihood of experiencing psychopathology and emotional distress. Our results point to the importance of being mindful and compassionate toward one’s own experiences and suggest that interventions that focus on the increasing ability to observe and accept one’s emotional experiences and being kind to oneself can be helpful for alleviating AjD and emotional distress symptoms, but more studies are needed.

## Figures and Tables

**Figure 1 healthcare-13-00934-f001:**
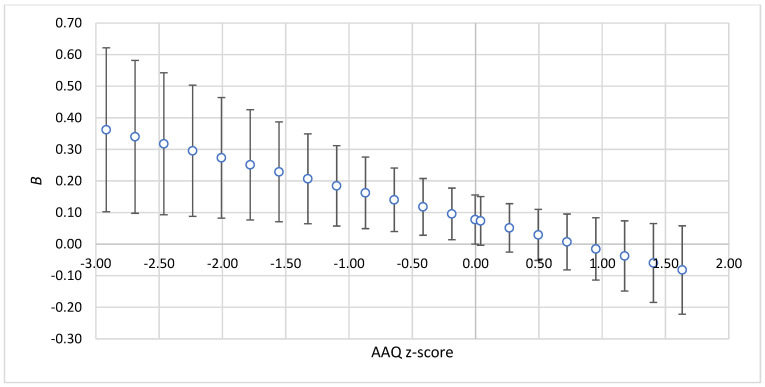
The strength of the relationship between PHLRC and ADNM-20 depending on the values of AAQ-II. Note: *z*-score—standardized value of moderator; *B*—standardized regression coefficient for the relationship between PHLRC and ADNM. The bars show 95%CIs for standardized regression coefficients.

**Figure 2 healthcare-13-00934-f002:**
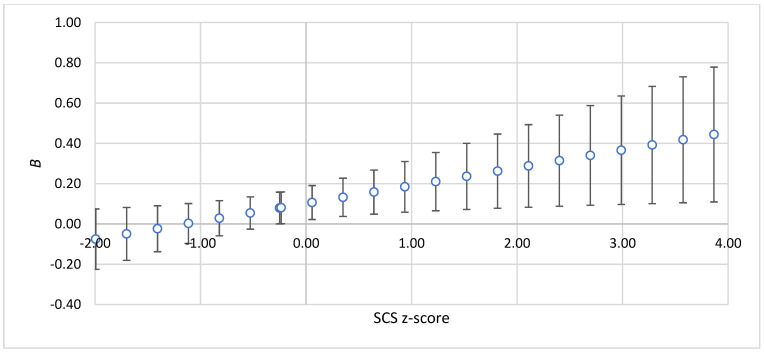
The strength of the relationship between PHLRC and ADNM-20 depending on the values of SCS. Note: *z*-score—standardized value of moderator; *B*—standardized regression coefficient for the relationship between PHLRC and ADNM. The bars show 95%CIs for standardized regression coefficients.

**Figure 3 healthcare-13-00934-f003:**
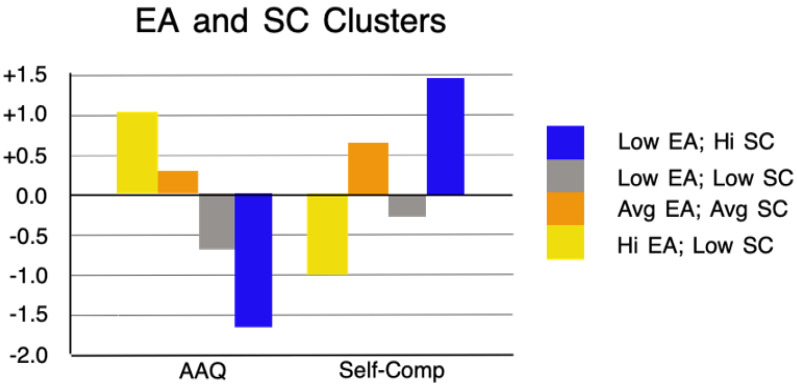
Final cluster centers, arithmetic means for standardized values of AAQ (EA), and self-compassion (SC) in the extracted clusters.

**Figure 4 healthcare-13-00934-f004:**
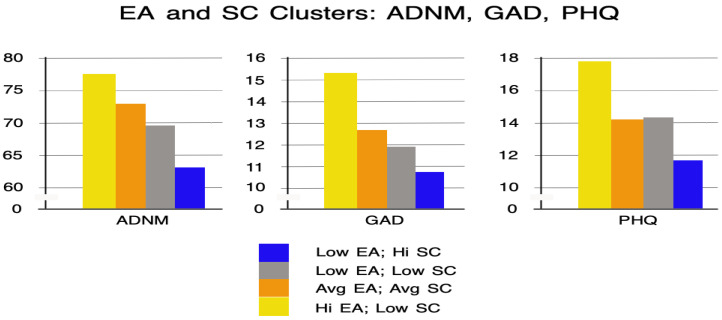
Mean values of PHQ, GAD, and ADNM intensity in the extracted four clusters.

**Table 1 healthcare-13-00934-t001:** Sociodemographic characteristics of the sample.

Gender	
Woman	281 (91.2%)
Man	21 (6.8%)
Other gender	6 (1.9%)
Age	18–61
Mean age	32.04 (SD = 9.40)
Education	
Elementary education	1 (0.3%)
Vocational education	1 (0.3%)
Secondary education	41 (13.3%)
Post-secondary education	50 (16.2%)
Higher education	215 (53.2%)

**Table 2 healthcare-13-00934-t002:** Descriptive statistics and Pearson correlation coefficients between analyzed variables.

	*M*	*SD*	1.	2.	3.	4.	5.	6.
1. PHLRC	19.11	4.54	-	-	-	-	-	-
2. ANDM-20	63.06	20.90	0.081	-	-	-	-	-
3. PHQ-9	15.02	5.34	0.125 *	0.489 **	-	-	-	-
4. GAD-7	13.05	4.49	0.150 **	0.452 **	0.642 **	-	-	-
5. HADS	21.70	7.00	0.105	0.656 **	0.579 **	0.545 **	-	-
6. AAQ-II	34.29	9.01	0.045	0.342 **	0.405 **	0.399 **	0.406 **	-
7. SCS_SF	2.19	0.60	−0.003	−0.253 **	−0.386 **	−0.351 **	−0.368 **	−0.568 **

Note: * *p* < 0.05; ** *p* < 0.01; *M*—mean value; *SD*—standard deviation.

**Table 3 healthcare-13-00934-t003:** Results of moderation analysis between PHLRC and ADNM-20.

Moderator	Predictors	95% *B*	*p*	*R* ^2^
AAQ	PHLRC	−0.01; 0.16	0.051	0.14
	AAQ	0.17; 0.32	0.001	
	PHLRC × AAQ	−0.18; −0.02	0.017	
Self-compassion	PHLRC	0.02; 0.18	0.017	0.09
	Self-compassion	−0.26; −0.10	0.001	
	PHLRC × Self-compassion	0.01; 0.17	0.025	

Note: 95% B—95% confidence interval for standardized regression coefficient; *p*—statistical significance; *R*^2^—determination coefficient.

**Table 4 healthcare-13-00934-t004:** The strength of the relationship between PHLRC and ADNM-20 depending on the values of moderators.

AAQ *z*-Score	*B*	*t*	*p*	SCS *z*-Score	*B*	*t*	*p*
−2.92	0.36	2.75	0.006	−1.99	−0.08	−0.99	0.323
−2.69	0.34	2.76	0.006	−1.70	−0.05	−0.74	0.458
−2.46	0.32	2.78	0.006	−1.41	−0.02	−0.41	0.683
−2.23	0.30	2.80	0.006	−1.11	0.00	0.05	0.964
−2.01	0.27	2.82	0.005	−0.82	0.03	0.64	0.523
−1.78	0.25	2.83	0.005	−0.53	0.05	1.34	0.183
−1.55	0.23	2.85	0.005	−0.25	0.08	1.97	0.050
−1.32	0.21	2.86	0.005	−0.24	0.08	2.00	0.047
−1.10	0.18	2.85	0.005	0.06	0.11	2.48	0.014
−0.87	0.16	2.82	0.005	0.35	0.13	2.74	0.007
−0.64	0.14	2.74	0.007	0.64	0.16	2.85	0.005
−0.41	0.12	2.58	0.010	0.94	0.18	2.87	0.004
−0.19	0.10	2.30	0.022	1.23	0.21	2.86	0.005
0.00	0.08	1.97	0.050	1.52	0.24	2.83	0.005
0.04	0.07	1.87	0.062	1.82	0.26	2.80	0.006
0.27	0.05	1.31	0.190	2.11	0.29	2.76	0.006
0.50	0.03	0.71	0.479	2.40	0.31	2.73	0.007
0.72	0.01	0.15	0.878	2.69	0.34	2.70	0.007
0.95	−0.02	−0.31	0.761	2.99	0.37	2.68	0.008
1.18	−0.04	−0.66	0.507	3.28	0.39	2.65	0.008
1.41	−0.06	−0.94	0.348	3.57	0.42	2.63	0.009
1.63	−0.08	−1.15	0.250	3.87	0.44	2.61	0.010

Note: *z*-score—standardized value of moderator; *B*—standardized regression coefficient for the relationship between PHLRC and ADNM; *t*—the value of the statistical test for regression coefficient; *p*—statistical significance.

**Table 5 healthcare-13-00934-t005:** Mean values of PHQ, GAD, and ADNM intensity in the extracted clusters.

	Cluster										
	Low EA, High SC		Average EA and SC		Low EA and SC		High EA, Low SC				
	*M*	*SD*	*M*	*SD*	*M*	*SD*	*M*	*SD*	*F*	*df*	*p*
ADNM	63.00	13.13	72.88	14.56	69.40	11.78	77.89	16.46	11.57	3,304	0.001
GAD	10.80	5.05	12.74	4.10	11.91	3.88	15.24	4.15	14.81	3,202	0.001
PHQ	11.70	5.41	14.23	4.67	14.31	4.93	17.82	5.01	17.75	3,303	0.001

Note: *M*—mean value; *SD*—standard deviation; *F*—analysis of variance statistics; *df*—degrees of freedom; *p*—statistical significance.

## Data Availability

Data are publicly available on the Open Science Framework at the following link (https://osf.io/zrxfe/files/osfstorage, accessed on 8 June 2020).

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
