# Peer review of "Disconnecting from Difficult Emotions in Times of Crisis: The Role of Self-Compassion and Experiential Avoidance in the Link Between Perceived COVID-19 Threat and Adjustment Disorder Severity"

_healthcare, 2025, doi:10.3390/healthcare13080934_

Round 1
Reviewer 1 Report
Comments and Suggestions for Authors
The research is interesting for the scientific community. After the COVID-19 pandemic, mental health cases have increased. The research conducts a study using the cluster analysis method to find out the relationship between the perceived threat of COVID-19 and the severity of adjustment disorder, examining self-compassion (SC) and experiential avoidance (EA) as potential moderators.
The Introduction section is interesting, but requires reinforcement from previous research to strengthen the statements made by the authors, such as: “Given the detrimental impact of perceived threat on stress response and coping mechanisms (Braun-Lewensohn 58 & Al-Sayed, 2018), it is a significant predictor of AjD. However, little research has explored 59 the moderators of the relationship between perceived threat and AjD symptoms”. The authors indicate that the detrimental impact of perceived threat on stress response and coping mechanisms is a predictor of bipolar disorder. In this regard, the authors only cite one study from 2018. More recent citations are required.
The sample is small (308) but representative when considering the sampling frame available and the set of key variables that are correlated with the variables of interest.
The analyses are basic but characteristic. The description of the analyses is well explained, allowing us to understand the interpretation of the data. The cluster analysis resembled established profiles in the SC and EA scores, clusters compared in the severity of the symptoms of AjD, PHQ and GAD.
The conclusions of the research are interesting. The authors highlight that individuals with low EA and high SC demonstrated significantly better psychological well-being. They present a fundamental basis for conducting research taking into account the findings of this research.
Author Response
Dear Reviewer 1,
Thank you for all the comments and suggestions to our manuscript " Disconnecting from Difficult Emotions in Times of Crisis: The Role of Self-Compassion and Experiential Avoidance in the Link Between Perceived COVID-19 Threat and Adjustment Disorder Severity". We would like to re-submit the improved versions of the manuscript to Healthcare. Below you will find our point-by-point replies to the Reviewers' comments. All the changes in the revised manuscript are marked with red font.
Reviewer: 1 - Comments to the Author
The research is interesting for the scientific community. After the COVID-19 pandemic, mental health cases have increased. The research conducts a study using the cluster analysis method to find out the relationship between the perceived threat of COVID-19 and the severity of adjustment disorder, examining self-compassion (SC) and experiential avoidance (EA) as potential moderators.
1. The Introduction section is interesting, but requires reinforcement from previous research to strengthen the statements made by the authors, such as: “Given the detrimental impact of perceived threat on stress response and coping mechanisms (Braun-Lewensohn 58 & Al-Sayed, 2018), it is a significant predictor of AjD. However, little research has explored 59 the moderators of the relationship between perceived threat and AjD symptoms”. The authors indicate that the detrimental impact of perceived threat on stress response and coping mechanisms is a predictor of bipolar disorder. In this regard, the authors only cite one study from 2018. More recent citations are required.
Reply: Thank you. We improved this section in the following way including also other, more recent citations. Note, however, that we describe Adjustment Disorder (AjD), not a bipolar disorder.
.. Given the detrimental impact of perceived threat on stress response and coping mechanisms [15], it is unsurprising that the perceived threat of the COVID-19 pandemic has emerged as a significant predictor of AjD. For instance, a Polish study during the pandemic's early phase found that 75% of participants considered COVID-19 a highly stressful event, which was the strongest predictor of AjD. Additionally, 49% reported increased AjD symptoms, with higher prevalence among females and those without full-time employment [16]. The European Society for Traumatic Stress Studies (ESTSS) conducted the ADJUST Study, involving 15,563 adults across eleven European countries, which found a prevalence of self-reported probable adjustment disorder (AjD) of 18.2% [17]. Pandemic-related stressors associated with higher levels of AjD symptoms included, among others, fear of infection. These findings underscore the important role of perceived threat in the development of AjD during the COVID-19 pandemic. However, little research has explored the moderators of the relationship between perceived threat and AjD symptoms.
[15] Braun-Lewensohn, O., Al-Sayed, K. (2018). Syrian adolescent refugees: How do they cope during their stay in refugee camps?. Frontiers in psychology, 9, 1258.
[16] Dragan, M., Grajewski, P., Shevlin, M. (2021). Adjustment disorder, traumatic stress, depression and anxiety in Poland during an early phase of the COVID-19 pandemic. European Journal of Psychotraumatology, 12(1), 1860356.
[17] Lotzin, A., Krause, L., Acquarini, E., Ajdukovic, D., Ardino, V., Arnberg, F., ... & ADJUST Study Consortium. (2021). Risk and protective factors, stressors, and symptoms of adjustment disorder during the COVID-19 pandemic–First results of the ESTSS COVID-19 pan-European ADJUST study. European journal of psychotraumatology, 12(1), 1964197.
The sample is small (308) but representative when considering the sampling frame available and the set of key variables that are correlated with the variables of interest.
The analyses are basic but characteristic. The description of the analyses is well explained, allowing us to understand the interpretation of the data. The cluster analysis resembled established profiles in the SC and EA scores, clusters compared in the severity of the symptoms of AjD, PHQ and GAD.
Thank you.
Reviewer 2 Report
Comments and Suggestions for Authors
I suggest that the authors consider the recommendations provided in order to enhance the manuscript's readability and make it more engaging. The topic, in fact, lacks novelty and would benefit from a revision to better capture the attention of the scientific community.
Review Report for Manuscript [Healthcare-3522287]
Title: Disconnecting from Difficult Emotions in Times of Crisis: The Role of Self-Compassion and Experiential Avoidance in the Link Between Perceived COVID-19 Threat and Adjustment Disorder Severity
The paper examines the relationship between perceived COVID-19 threat and the severity of adjustment disorder (AjD), investigating self-compassion (SC) and experiential avoidance (EA) as potential moderators. This addresses a significant issue in the field of mental health, which has been notably impacted worldwide by the COVID-19 pandemic. The authors explore how participants in the study were able, or unable, to protect their psychological well-being in the face of concerns related to the pandemic.
The topic is relevant to trends examining the repercussions of the COVID-19 pandemic.
The writing is generally clear and well-organized, with a logical flow of ideas that follows a coherent structure. The citations are appropriately integrated, providing strong contextual background for the research and helping to establish a solid foundation for the study.
The reading of the article, however, encourages me to offer several suggestions for improvement
- Introduction: The introduction is well-written; however, with a few adjustments, it could be made clearer and more precise for an international audience. Some passages are somewhat lengthy and could benefit from condensation to maintain focus. For instance, the repeated use of "AjD" and "perceived threat" leads to redundancy, which makes the text heavier than necessary. Additionally, it would be helpful to provide a clearer explanation of the term "cluster" to assist readers who may not be familiar with this terminology. Furthermore, a brief explanation of the concept of "moderation" in the psychological context would improve comprehension
- 2. Materials and Methods: The phrase "the invitation to the study was sent" could be revised to “invitations to the study were sent” to maintain consistency with the plural form. Additionally, other minor grammatical adjustments could improve clarity. For example, “Acceptance of informed consent was mandatory to take part in the study” could be rewritten as “Informed consent was mandatory for participation in the study” for greater precision.
- 2.3: Ensure that all information regarding scores and test reliability is presented with equal clarity. For instance, in the Anxiety section (GAD-7), the test-retest reliability should be explicitly stated.
- 2.4: Consider restructuring certain sentences to enhance clarity and fluency, particularly in the descriptions of moderation and cluster analysis, in order to avoid overly long and complex sentences. Additionally, abbreviations such as PHLRC and ADNM-20 should be spelled out upon their first occurrence for clarity. The citation style should be adjusted (e.g., “Hayes, 2017”) to align with standardized academic writing conventions. Greater precision is needed in describing the moderation test and cluster analysis, ensuring that there is no ambiguity in the sections detailing the statistical models and their applications.
- Results: The phrasing regarding the correlation between PHLRC and the symptom intensity scales could be improved to enhance readability and fluidity.
- Discussion: Enhance the fluency of certain sentences and ensure grammatical consistency throughout. Some terms could be replaced with more accurate alternatives (e.g., “emotional avoidance” could be substituted with “experiential avoidance,” which is a more precise term in psychology). The use of logical connectors would improve the flow of the discussion, and punctuation should be consistent. Furthermore, the citation format should be uniform throughout the text (e.g., “Kroenke et al., 2001”).
Finally, I recommend incorporating more recent studies on this topic to provide a more up-to-date context:
Ruggieri,S., Ingoglia, S., Bonfanti, R.C., and Lo Coco, G. (2021).The role of online social comparison as a protective factor for psychological wellbeing: A longitudinal study during the COVID-19 quarantine. Personality and Individual Differences, 171, 110486. https://doi.org/10.1016/j.paid.2020.110486.
World Health Organization. (2020). Considerations for quarantine of individuals in the context of containment for coronavirus disease (COVID-19). Retrieved from https ://apps.who.int/iris/bitstream/handle/10665/331497/WHO-2019-nCoVIHR_Quarantine-2020.2- eng.pdf.
The paper demonstrates significant potential but requires several revisions before it can be considered for publication.
I recommend a major revision of the manuscript.
Comments on the Quality of English LanguageLanguage supervision is recommended
Author Response
Dear Reviewer 2,
Thank you for all the comments and suggestions to our manuscript " Disconnecting from Difficult Emotions in Times of Crisis: The Role of Self-Compassion and Experiential Avoidance in the Link Between Perceived COVID-19 Threat and Adjustment Disorder Severity". We would like to re-submit the improved versions of the manuscript to Healthcare. Below you will find our point-by-point replies to the Reviewers' comments. All the changes in the revised manuscript are marked with red font.
Reviewer: 2 - Comments to the Author
Introduction:
Comment: The introduction is well-written; however, with a few adjustments, it could be made clearer and more precise for an international audience. Some passages are somewhat lengthy and could benefit from condensation to maintain focus. For instance, the repeated use of "AjD" and "perceived threat" leads to redundancy, which makes the text heavier than necessary. Additionally, it would be helpful to provide a clearer explanation of the term "cluster" to assist readers who may not be familiar with this terminology. Furthermore, a brief explanation of the concept of "moderation" in the psychological context would improve comprehension
Reply: We appreciate these thoughtful comments. Clarity is indeed essential, and in response, we have revised the manuscript accordingly - starting from the Abstract, as suggested (see also replies for other comments).
An abstract: Additionally, cluster analysis—a statistical method for grouping individuals based on similar psychological characteristics—was employed to identify distinct profiles of SC and EA and their associations with AjD, depression, and anxiety symptoms.
In introduction:
“Both SC and EA have been shown to play crucial roles in emotional regulation, and in this study, we explored whether they serve as moderators—variables that influence the strength or direction of the relationship—between perceived threat of COVID-19 (hereafter referred to as 'perceived threat') and AjD symptoms.”
Materials and Methods:
Comment: The phrase "the invitation to the study was sent" could be revised to “invitations to the study were sent” to maintain consistency with the plural form. Additionally, other minor grammatical adjustments could improve clarity. For example, “Acceptance of informed consent was mandatory to take part in the study” could be rewritten as “Informed consent was mandatory for participation in the study” for greater precision.
Reply: Thank you very much for these corrections. English is not my mother of tongue.
Comment: 2.3: Ensure that all information regarding scores and test reliability is presented with equal clarity. For instance, in the Anxiety section (GAD-7), the test-retest reliability should be explicitly stated.
Reply. We correct it (below). See also modifications for other measures.
“The GAD-7 demonstrated excellent psychometric properties. Internal consistency was high (Cronbach’s α = .92), and test-retest reliability, assessed via intraclass correlation, was also strong (ICC = .83).”
Comment: 2.4: Consider restructuring certain sentences to enhance clarity and fluency, particularly in the descriptions of moderation and cluster analysis, in order to avoid overly long and complex sentences. Additionally, abbreviations such as PHLRC and ADNM-20 should be spelled out upon their first occurrence for clarity. The citation style should be adjusted (e.g., “Hayes, 2017”) to align with standardized academic writing conventions. Greater precision is needed in describing the moderation test and cluster analysis, ensuring that there is no ambiguity in the sections detailing the statistical models and their applications.
Reply: Thank you. We spelled out these abbreviations and adjusted the citation style.
In addition, we put more emphasis on the clarity of description. We modified almost the whole section of statistical analysis to increase it, as follows:
“We began by calculating descriptive statistics and Pearson correlation coefficients among the study variables. Moderation analyses were then conducted to examine whether experiential avoidance (EA) and self-compassion (SC) influenced the strength of the relationship between perceived COVID-19 threat (PHLRC) and adjustment disorder severity (ADNM-20). These analyses were carried out using Hayes' PROCESS macro for SPSS (version 3.5.3), employing Model 1 to test individual moderation effects and Model 3 to test for potential interaction between moderators [40]. All predictors were mean-centered prior to analysis to facilitate interpretation. Statistical significance of moderation was assessed through 95% confidence intervals (CI) derived via bootstrapping (5000 samples). The Johnson-Neyman technique was applied to identify regions of significance for the interaction effects.”
… “In Model 3, we evaluated the three-way interaction between PHLRC, SC, and EA to determine whether the moderating effects of SC and EA were conditional on one another. The Johnson-Neyman procedure was applied to identify the range of moderator values where the conditional effect of the predictor (PHLRC) on the outcome (ADNM-20) was statistically significant. Results were visualized with interaction plots and interpreted based on standardized coefficients.”
“To examine heterogeneity in psychological profiles, a k-means cluster analysis was conducted using standardized scores of AAQ-II (EA) and SCS-SF (SC) as input variables. The analysis was performed using IBM SPSS Statistics 28.0. A four-cluster solution was chosen based on theoretical assumptions and supported by visual inspection of the within-cluster sum of squares. The goal was to classify participants into distinct psychological profiles representing combinations of high/low SC and EA. The final cluster solution comprised: 1. High SC / Low EA; 2. Average SC / Average EA; 3. Low SC / Low EA; 4. Low SC / High EA.”
.. “Following cluster extraction, a one-way ANOVA was conducted to compare levels of AjD, depression (PHQ-9), and anxiety (GAD-7) symptoms across the four groups. Post-hoc comparisons were conducted using the Games-Howell test to account for heterogeneity of variance. Effect sizes (η²) were computed for each outcome to assess the magnitude of between-cluster differences.”
Results:
Comment: The phrasing regarding the correlation between PHLRC and the symptom intensity scales could be improved to enhance readability and fluidity.
Reply: We modified it as follows:
“A positive correlation was observed between perceived health risk (PHLRC) and both depressive and anxiety symptoms, suggesting that individuals perceiving greater threat also reported more intense psychological distress.”
Discussion:
Comment: Enhance the fluency of certain sentences and ensure grammatical consistency throughout. Some terms could be replaced with more accurate alternatives (e.g., “emotional avoidance” could be substituted with “experiential avoidance,” which is a more precise term in psychology). The use of logical connectors would improve the flow of the discussion, and punctuation should be consistent. Furthermore, the citation format should be uniform throughout the text (e.g., “Kroenke et al., 2001”).
Reply: Thank you for this comment. We addressed the issue of consistency and logical connectors in numerous places within the discussion, and we have also abbreviated it.
For example:
“Importantly, the moderating effect tells us nothing about the actual level of the dependent measure—in our case, AjD symptoms—and it is worth noting that participants in the present study were selected based on meeting diagnostic criteria for AjD (ADNM-20) and elevated emotional distress (anxiety and depression) on the HADS. Therefore, it is possible that these individuals tended to disconnect from their emotional experiences and were harsh, rather than compassionate, toward their difficulties to prevent emotional distress triggered by worries about their own and their loved ones’ health and safety. As a result, EA and a lack of SC may have prevented emotional elaboration of pandemic-related worries, thereby helping participants to reduce the adverse emotional impact—at least temporarily. In the short term, this strategy may have "protected" them from heightened AjD symptoms during the pandemic.”
“A crucial question, however, is whether this protective function led to improved psychological functioning. In other words, did individuals with high EA and low SC exhibit reduced levels of depression, anxiety, and AjD symptoms compared to those with opposite characteristics (low EA and high SC)? To address this, we conducted a cluster analysis to identify participant profiles based on SCS-SF and AAQ scores and compared these profiles across symptom severity. The results were clear. Specifically, individuals with high psychological flexibility and high SC exhibited significantly lower levels of AjD, depression, and anxiety symptoms compared with those with high EA and low SC. This suggests that, although avoidance and self-criticism might initially dampen emotional reactions, over time, they are associated with increased psychological distress.”
“Generally speaking, our findings indicate that individuals may use experiential avoidance and harsh self-judgment in an attempt to minimize distress. However, these strategies may eventually lead to worse outcomes. This interpretation is supported by previous experimental studies that found similar roles for SC and EA in the development of psychopathology [31]. Furthermore, our findings are in line with clinical observations suggesting that individuals with poor emotional insight and low self-reflection often exhibit high symptom levels and poor awareness of emotional causes [46].”
Comment: Finally, I recommend incorporating more recent studies on this topic to provide a more up-to-date context:
Ruggieri,S., Ingoglia, S., Bonfanti, R.C., and Lo Coco, G. (2021).The role of online social comparison as a protective factor for psychological wellbeing: A longitudinal study during the COVID-19 quarantine. Personality and Individual Differences, 171, 110486. https://doi.org/10.1016/j.paid.2020.110486.
World Health Organization. (2020). Considerations for quarantine of individuals in the context of containment for coronavirus disease (COVID-19). Retrieved from https ://apps.who.int/iris/bitstream/handle/10665/331497/WHO-2019-nCoVIHR_Quarantine-2020.2- eng.pdf.
Reply: We accommodate those two references as follows:
The shifting threat landscape, uncertain health and economic conditions, and widespread disruption of daily life created conditions that mirror the diagnostic profile of AjD—namely, a preoccupation with the stressor and significant difficulty adjusting to it. At the same time, certain behavioral and cognitive strategies, such as adaptive social comparisons or online engagement, may have helped buffer psychological well-being during lockdowns [10]. Therefore, AjD may offer a uniquely relevant lens for understanding mental health vulnerability during the COVID-19 crisis.”
“Factors such as self-isolation, quarantine, job loss, and perceived risk of contracting COVID-19 were identified as key risk factors for developing AjD [8]. Indeed, international health guidelines recognized quarantine as a potentially distressing experience with psychological consequences, particularly if prolonged [9].”
[8] Lotzin, A., Acquarini, E., Ajdukovic, D., Ardino, V., Böttche, M., Bondjers, K., Schäfer, I. (2020). Stressors, coping and symptoms of adjustment disorder in the course of the COVID-19 pandemic–study protocol of the European Society for Traumatic Stress Studies (ESTSS) pan-European study. European journal of psychotraumatology, 11(1), 1780832.
[9] World Health Organization. (2020). Considerations for quarantine of individuals in the context of containment for coronavirus disease (COVID-19). Retrieved from https ://apps.who.int/iris/bitstream/handle/10665/331497/WHO-2019-nCoVIHR_Quarantine-2020.2- eng.pdf.
[10] Ruggieri,S., Ingoglia, S., Bonfanti, R.C., and Lo Coco, G. (2021).The role of online social comparison as a protective factor for psychological wellbeing: A longitudinal study during the COVID-19 quarantine. Personality and Individual Differences, 171, 110486. https://doi.org/10.1016/j.paid.2020.110486.
We also added references to more recent studies, including:
[16] Dragan, M., Grajewski, P., Shevlin, M. (2021). Adjustment disorder, traumatic stress, depression and anxiety in Poland during an early phase of the COVID-19 pandemic. European Journal of Psychotraumatology, 12(1), 1860356.
[17] Lotzin, A., Krause, L., Acquarini, E., Ajdukovic, D., Ardino, V., Arnberg, F., ... & ADJUST Study Consortium. (2021). Risk and protective factors, stressors, and symptoms of adjustment disorder during the COVID-19 pandemic–First results of the ESTSS COVID-19 pan-European ADJUST study. European journal of psychotraumatology, 12(1), 1964197.
Reviewer 3 Report
Comments and Suggestions for Authors
This is a well written paper with the potential to add the literature on Adjustment Disorder Severity at a specific level, but within the broader fields of depression and anxiety. Authors can be commended for the robust aspects of the methodology and for the focus on the experiences within the COVID-19 period to leverage on the lessons learnt and insights to take forward. The discussion is also sound with clear engagement of the findings, but with some unpacking as well of the questions that unfolded through this process. There are some recommendations for minor improvements in the paper. These are as follows:
- Introduction-The situating of adjustment disorder seems limited and can be expanded linking this to the concerns that emerged within the pandemic. This addition will strengthen the justification for the paper from the onset.
- Statistical testing and reporting of variations in adjustment disorder severity based on the socio-demographic background of participants. This is presented in the description of the sample but not utilized within the exploration of the data.
- Discussion-Excellent discussion here generally. Two recommendations however would be to add more on the demographic findings once reported to compare with other studies in the field and to add a section on the implications of the research for theory, research and practice.
Author Response
Dear Reviewer 3,
Thank you for all the comments and suggestions to our manuscript " Disconnecting from Difficult Emotions in Times of Crisis: The Role of Self-Compassion and Experiential Avoidance in the Link Between Perceived COVID-19 Threat and Adjustment Disorder Severity". We would like to re-submit the improved versions of the manuscript to Healthcare. Below you will find our point-by-point replies to the Reviewers' comments. All the changes in the revised manuscript are marked with red font.
Comment: This is a well written paper with the potential to add the literature on Adjustment Disorder Severity at a specific level, but within the broader fields of depression and anxiety. Authors can be commended for the robust aspects of the methodology and for the focus on the experiences within the COVID-19 period to leverage on the lessons learnt and insights to take forward. The discussion is also sound with clear engagement of the findings, but with some unpacking as well of the questions that unfolded through this process. There are some recommendations for minor improvements in the paper. These are as follows:
Introduction-The situating of adjustment disorder seems limited and can be expanded linking this to the concerns that emerged within the pandemic. This addition will strengthen the justification for the paper from the onset.
Reply: Thank you for this suggestion, we added the following paraghraph in the Introduction:
“Given the pandemic's global and prolonged nature, it represented a unique type of stressor—ubiquitous, ongoing, and often ambiguous—which aligns closely with the kinds of stressors that trigger AjD. While much research has focused on PTSD, anxiety, and depression during the pandemic, adjustment disorder offers a more con-text-specific framework for understanding short- to mid-term psychological maladaptation in response to such diffuse life changes. The shifting threat landscape, uncertain health and economic conditions, and widespread disruption of daily life created conditions that mirror the diagnostic profile of AjD—namely, a preoccupation with the stressor and significant difficulty adjusting to it. At the same time, certain behavioral and cognitive strategies, such as adaptive social comparisons or online engagement, may have helped buffer psychological well-being during lockdowns [10]. Therefore, AjD may offer a uniquely relevant lens for understanding mental health vulnerability during the COVID-19 crisis.”
Comment: Statistical testing and reporting of variations in adjustment disorder severity based on the socio-demographic background of participants. This is presented in the description of the sample but not utilized within the exploration of the data.
Reply: Thank you for this comment, we appreciate it.
We added the following paragraph in the results section:
Regarding sociodemographic variables, we refrained from analyzing gender differences due to the overrepresentation of female participants. No significant correlation was found between age and the intensity of ADNM symptoms, r(306) = .02, p > .05. Similarly, participants with higher education did not differ significantly in ADNM symptom severity compared to those without higher education, t(306) = 0.98, p > .05.
We also inserted the following section into the discussion:
Previous research has indicated that female gender and younger age are associated with higher levels of AjD symptoms [4, 16–17]. Although the overrepresentation of female participants in our sample limited the feasibility of meaningful gender comparisons, we conducted an analysis to examine the relationship between age and AjD severity. However, no significant association was observed between these variables.
We also added the following limitation of the study:
“In addition, the sample was characterized by a gender imbalance, with a majority of participants identifying as women. This uneven distribution may limit the generalizability of the findings, particularly in terms of their applicability to male or gender-diverse populations. Future studies should aim to recruit more gender-balanced samples or focus specifically on one gender to explore potential differences in psychological responses.”
Comment: Discussion -Excellent discussion here generally. Two recommendations however would be to add more on the demographic findings once reported to compare with other studies in the field and to add a section on the implications of the research for theory, research and practice.
Reply: Thank you for these valuable suggestions. We have substantially revised the Discussion section, and we hope that the current version adequately addresses the issues raised.
We also added also a section on the implications of the research, as follows:
“Implication of the research for theory, research and practice
This study offers several key implications for psychological theory, future research, and clinical practice.
Theoretical Implications: Our findings support theoretical models within third-wave cognitive-behavioral therapies that emphasize the importance of mindfulness, experiential acceptance, and self-compassion in emotional regulation. The results reinforce the theoretical relevance of combining emotional avoidance and compassionate self-awareness frameworks in understanding psychopathological outcomes during prolonged stress.
Research Implications: The study highlights the need for further longitudinal and experimental research to examine how these constructs interact over time, especially during ongoing global crises. Future studies should explore the temporal dynamics of SC and EA and their potential causal relationships with mental health outcomes. Additionally, the use of real-time assessment methods such as ecological momentary assessment could provide a more nuanced understanding of how these processes unfold in daily life. Expanding samples to include diverse populations and gender-balanced groups will enhance the generalizability of findings.
Practical Implications: Clinically, these findings underscore the importance of interventions aimed at enhancing self-compassion and reducing experiential avoidance in individuals experiencing adjustment difficulties. Therapeutic approaches such as Mindfulness-Based Cognitive Therapy (MBCT) and Acceptance and Commitment Therapy (ACT) that foster psychological flexibility and compassionate self-relating may be particularly beneficial in mitigating the emotional impact of perceived threats. Promoting emotional openness and self-kindness may reduce vulnerability to AjD, especially in times of collective crisis like a pandemic).“
Round 2
Reviewer 2 Report
Comments and Suggestions for Authors
In my view, the manuscript is currently available for publication as the authors implemented the suggestions received after the first revision.